# Variable Speed Limit and Ramp Metering for Mixed Traffic Flows: A Review and Open Questions

Filip Vrbanić *,† , Edouard Ivanjko † , Krešimir Kušić and Dino Čakija

Faculty of Transport and Traffic Sciences, University of Zagreb, Vukelićeva Street 4, HR-10000 Zagreb, Croatia; edouard.ivanjko@fpz.unizg.hr (E.I.); kresimir.kusic@fpz.unizg.hr (K.K.); dino.cakija@fpz.unizg.hr (D.Č.)
* Correspondence: filip.vrbanic@fpz.unizg.hr
† These authors contributed equally to this work.

**Abstract:** The trend of increasing traffic demand is causing congestion on existing urban roads, including urban motorways, resulting in a decrease in Level of Service (LoS) and safety, and an increase in fuel consumption. Lack of space and non-compliance with cities' sustainable urban plans prevent the expansion of new transport infrastructure in some urban areas. To alleviate the aforementioned problems, appropriate solutions come from the domain of Intelligent Transportation Systems by implementing traffic control services. Those services include Variable Speed Limit (VSL) and Ramp Metering (RM) for urban motorways. VSL reduces the speed of incoming vehicles to a bottleneck area, and RM limits the inflow through on-ramps. In addition, with the increasing development of Autonomous Vehicles (AVs) and Connected AVs (CAVs), new opportunities for traffic control are emerging. VSL and RM can reduce traffic congestion on urban motorways, especially so in the case of mixed traffic flows where AVs and CAVs can fully comply with the control system output. Currently, there is no existing overview of control algorithms and applications for VSL and RM in mixed traffic flows. Therefore, we present a comprehensive survey of VSL and RM control algorithms including the most recent reinforcement learning-based approaches. Best practices for mixed traffic flow control are summarized and new viewpoints and future research directions are presented, including an overview of the currently open research questions.

**Keywords:** intelligent transportation systems; urban motorways; traffic control; ramp metering; variable speed limit; connected and autonomous vehicles; reinforcement learning

## 1. Introduction

The ever-increasing traffic demand, especially on urban roads and motorways, leads to occasional under-capacity on individual road sections. This effect is also significant on urban motorways—a type of motorway located near large urban areas. Their primary purpose is to connect the urban area with smaller industrial, residential or rural areas. Urban motorways include a series of on-ramps and off-ramps that connect the urban motorway to the local urban road network. Ramps on urban motorways are relatively close to each other in contrast to rural motorways outside urban areas. The increase in traffic demand under such conditions leads to a decrease in the Level of Service (LoS) of the motorway. This is described by lower speed, higher traffic density and longer Travel Time ($TT$). Traffic disturbances or congestion usually occurs in the motorway's on-ramp merging areas due to the increased inflow. This inflow causes a slowdown of the main traffic flow due to the merging of vehicles from the on-ramp into the mainline. Such phenomena are specific to peak demands or rush hour conditions. When the mainline traffic flow volume on a particular segment of the motorway exceeds the designed throughput (capacity), the so-called bottleneck occurs and the traffic flow becomes unstable. In this case, the bottleneck refers to an area of the urban motorway where the congestion occurs.

Due to increased traffic demand, the mainline traffic flow becomes unstable, resulting in more pronounced interactions between vehicles. Minor acceleration or deceleration of a

single vehicle or a group of vehicles leads to a significant change in adjacent vehicles' speed during unstable traffic flow conditions. Such a change in speed is known as a shock wave, which propagates upstream [1]. This is a traveling disturbance and causes new significant irregularities in the upstream traffic flow. Such a shock wave is one of the the main causes of additional capacity drop and congestion on a critical segment of urban motorway due to unstable traffic flow. At the time of capacity drop, the vehicle flow measured immediately after the bottleneck area is significantly reduced relative to the maximum possible capacity of the observed segment [2–4].

Traffic control is one of the services from the domain of Intelligent Transportation Systems (ITS) that can alleviate congestion. The main Measures of Effectiveness (MoEs) used to assess the performance of traffic control systems are $TT$, measured in units of time, e.g., (s), Total Travel Time ($TTT$), and the Total Time Spent ($TTS$) of all vehicles on the controlled segment of the urban motorway, both expressed in (veh · h). LoS is another measure used to describe the traffic state labeled with letters from $A$ to $F$ and described by density and speed measurements, according to Highway capacity manual [5]. On urban motorways, the most commonly used traffic control systems are Variable Speed Limit (VSL) and Ramp Metering (RM). VSL controls the speed of the main traffic flow on the motorway and thus affects the dynamics of the traffic flow. By changing the speed limit on Variable Message Signs (VMS), VSL indirectly controls the inflow of vehicles on the controlled segment [6]. Thus, the VSL control system aims to achieve higher operational capacity of the existing traffic infrastructure on urban motorways without the need for additional traffic lanes.

On the other hand, RM limits the number of vehicles entering the motorway from the controlled on-ramp. RM can be implemented as an open- or closed-loop control system. The main difference between open-loop and feedback systems is that the open-loop systems do not account the output of the system in the next control step, while feedback systems do. Classically used algorithms for RM use a feedback control loop such as the ALINEA algorithm.

VSL and RM can be integrated and act in synergy. In this case, VSL usually serves as a complement to RM, when a stand-alone RM system is unable to achieve predefined traffic parameter values (usually vehicle density or flow) within desired limits on a controlled segment of an urban motorway [7]. Studies [2,7,8] have shown that the VSL system can prevent or reduce the effect of capacity drop caused by increased traffic demand on the controlled urban motorway segment. By reducing the speed limit posted on the VMSs placed in front of the bottleneck area, the effect of reducing the vehicle inflow to the bottleneck area is achieved.

Today's traffic control systems increasingly rely on intelligent data processing. For example, the City Brain system is an urban traffic control platform that includes various traffic control systems alongside a large number of sensors for traffic data collection. It is based on the elastic computation and large-scale data processing platform of Alibaba Cloud, integrated with the capabilities of interdisciplinary fields such as machine vision, large-scale topological network computation, and traffic flow analysis [9]. It is capable of collecting massive multi-source data, real-time processing, and intelligent computing. It incorporates extremely large-scale and multi-source data processing, Machine Learning (ML), and formulating global optimal traffic control strategy.

Emerging technologies in Autonomous Vehicles (AVs) and Connected and Autonomous Vehicles (CAVs) are likely to improve the way VSL and RM will operate in the coming years. CAVs can be incorporated with VSL so they automatically comply with the posted speed limits. In the initial phase of deployment, AVs and CAVs will cohabit with Human-Driven Vehicles (HDVs) in mixed traffic flows. The term mixed traffic flows thus refers to traffic flows that contain conventional HDVs, AVs and CAVs with different penetration rates. AVs represent vehicles that can receive traffic information from different proprioceptive sensing technologies integrated within the vehicle such as cameras, sonars, navigation, radar, and lidar. They are characterized by high compliance with traffic laws, shorter time

and space headway, and lower gap acceptance. CAVs are similar to AVs with additional features that enable communication with other vehicles (Vehicle-to-Vehicle (V2V)), roadside infrastructure (Vehicle-to-Roadside (V2R) communication) such as traffic signal controllers, infrastructure (Vehicle-to-Infrastructure (V2I) communication), and the entire environment (Vehicle-to-Everything (V2X) communication) [10,11]. RM strategies in mixed traffic flows disregard the classical RM approaches, as traffic lights become obsolete for signaling the entry of vehicles onto the mainline flow. Instead of the classical RM control, which uses a traffic light with real-time traffic data to determine the rate at which vehicles should enter the motorway, merging control approaches are proposed to resolve interactions and conflicts between vehicles at merge areas. Thus, we refer to RM for HDV traffic flows and merging control approaches for mixed traffic flows. Therefore, for mixed traffic flows, merging control optimization is developed instead of RM.

This paper describes the VSL problem and the merging problem on urban motorways with a focus on mixed traffic flow scenarios. To the best of our knowledge, there is no comprehensive systematic review on applications of control strategies for VSL and merging approaches to mixed traffic flows. A brief overview of VSL and RM control strategies for HDV traffic flows is also mentioned to explain the current knowledge base for possible control strategies that can be applied to mixed traffic flows. A more comprehensive review of VSL strategies for HDV flows has been analyzed in [12–16], while in [17] the focus has been on Reinforcement Learning (RL) strategies for VSL control. RM strategies in HDV flows were analyzed in papers [18–26]. RL strategies for RM control were analyzed in [25]. The main motivation for this research is to present the state of the art of the currently used control strategies for solving the VSL and RM (merging control) approaches in mixed traffic flows on urban motorways.

Therefore, the main contributions of this study are:

- In this study, we applied the systematic literature review approach. We used a keyword-based search and systematically identified existing highly relevant studies from the search results.
- This study covers traffic control studies on urban motorways focused on VSL and merging control approaches in mixed traffic flows.
- First, we identified and categorized the control objectives, e.g., improving efficiency or safety, for scenarios with HDV flows. Then, we analyzed different approaches for VSL and merging control in mixed traffic flows. In addition, we identified and summarized papers that analyze the impact of mixed traffic flows on the fundamental diagram without control strategies. Finally, we categorized the main objectives of control approaches in mixed traffic flows.

The rest of the paper is organized as follows. Section 2 presents the application of VSL and RM control in HDV traffic flows using classical and RL-based approaches. Section 3 presents the impact of mixed traffic flows on the fundamental diagram and the applied control strategies for VSL and RM in mixed traffic flows. Section 4 discusses the presented VSL and RM control strategies in mixed traffic flows including their advantages and disadvantages. Section 5 presents the conclusion and suggestions for future work.

## 2. Application of VSL and RM Control in HDV Traffic Flows

The VSL control system application with timely and appropriate speed limits can control the incoming flow of the vehicles approaching the bottleneck by adjusting it to a lower level, i.e., to the amount of capacity of the bottleneck [6]. This prevents the capacity from dropping further and allows the congestion to be resolved more quickly and achieve traffic flow values less than or equal to the maximum capacity in the bottleneck area. The preventive effect of VSL is manifested in delaying the occurrence of the capacity drop, which prevents the activation of bottlenecks by timely control of the incoming traffic flow. The application of VSL leads to smaller speed deviations between vehicles and lanes, thus reducing the risk of accidents [27–30].

RM systems control the inflow of on-ramp motorway vehicles. For this purpose, the feedback system detectors required for operation are usually placed downstream of the on-ramp where vehicle merging takes place [31]. The most common RM algorithm, ALINEA [32], uses the output of the previous control step interval as input for the current iteration. Thus, the ALINEA inputs used are the previous control step interval metering rate and downstream occupancy. Some of the extensions to ALINEA that have been developed to improve performance are FL-ALINEA, UP-ALINEA, UF-ALINEA, AD-ALINEA, X-ALINEA/Q, and PI-ALINEA [19].

In the following subsections, the VSL and RM algorithms for HDV flows are analyzed, and grouped by the applied processing methods (rule-based, open-loop, feedback loop, and RL). Different metrics have been used as indicators of algorithm performance in different simulation scenarios. Thus, it is difficult to say which is the best method without in-depth simulation analysis using the same scenario, which is out of the scope of this paper—especially since each of them has a different optimization objective, such as minimizing emissions, improving safety, and improving chosen macroscopic traffic parameters. Thus, only a comment about the effectiveness of the respective method group is given using the metric results published in the surveyed literature.

### 2.1. Classical VSL Approaches

In this subsection, existing approaches for VSL implementation on urban motorways are presented. These classical approaches were developed for HDV-based traffic flows and can potentially be applied to (C)AVs if (C)AVs are used as sensors or actuators for VSL.

### 2.1.1. Rule-Based Reactive VSL

This category of VSL systems bases its logic for calculating speed limits on predefined thresholds for a given traffic flow state. Based on the measured values of traffic parameters (flow $q$ expressed in veh/h, density $\rho$ expressed in veh/km, or mean speed $\overline{v}$ expressed in km/h) on a given motorway segment, a predefined speed limit is activated, as shown in Figure 1 [33–36]. VSL activation can be based on measured flow rate or standard deviations of density, as analyzed in [37]. Another approach is a reactive VSL system with the aim of informing the drivers about a possible disruption in the traffic flow (traffic accident, congestion). As a result, the speed of the incoming vehicle flow is gradually reduced and adapted to the new traffic situation. Such VSL systems can also have a weather-based activation logic (related to fog, ice, strong wind, etc.) [38]. The final decision on the speed limit output is made by the operations staff of the traffic control center based on the proposed speed limit and the assessment of the traffic situation gained from experience.

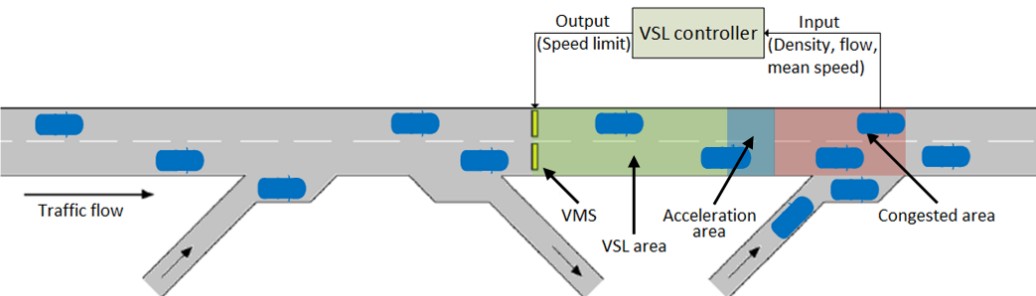

**Figure 1.** Block scheme of Variable Speed Limit (VSL) control on an urban motorway.

### 2.1.2. Open-Loop Based VSL

This category includes VSL controllers whose logic is based on the open-loop optimization process [7,39–41]. A non-feedback control system requires an accurate traffic model with the ability to predict the movement of traffic flow parameters on a given motorway segment. Such a predictive model is difficult to achieve due to the stochastic traffic behavior. Most often, predictive models are based on a general macroscopic model of

traffic flow. Other approaches, such as the Kalman filter [41], are used for better prediction of traffic flow to improve the optimization process.

### 2.1.3. VSL Based on Negative Feedback Loop

The strategy of feedback VSL controllers is also based on collecting current traffic data on the congested motorway segment as shown in Figure 1, but with the objective of maintaining the controlled motorway segment at a defined set point such as the critical density $\rho_c$. Such strategies do not require predictive traffic flow models. In [42], a simple VSL with integration action (so-called I-controller) for VSL control was proposed. It controls the flow of vehicles on the motorway segment before the problematic bottleneck occurence area. The working principle of the controller is based on the ALINEA algorithm [32]. Instead of a traffic light at the on-ramp controlling the flow of vehicles entering the urban motorway, VSL reduces the mainline speed limit. Similar to RM, VSL also controls the flow of vehicles by reducing the mainline speed. This achieves the effect of a "virtual ramp" on the motorway segment under VSL, which controls the output flow of the controlled segment or the inflow to the next adjacent problematic segment of the urban motorway. In [43], a distributed VSL controller based on control feedback adapted to eliminate shock waves was tested. The VSL on each controlled segment is operated by a separate controller (cumulatively 10 controllers). Such a VSL control system has proven to be successful in preventing the generation of shock waves, which cause minor disruptions to traffic flow. By applying VSL, $TTS$ was reduced by 20% compared to the uncontrolled case in [43]. In [44], a comprehensive feedback control strategy was proposed for mainline traffic flow control enabled via VSLs, considering multiple bottleneck locations. The feedback control results were compared with the optimal control results to evaluate the performance of the proposed strategy, and it was able to come close to the optimal control results.

### 2.2. RL-Based VSL

In [45], the Q-Learning (QL) algorithm was applied to VSL to optimize traffic flow on motorways taking traffic predictions into account. The environment was described by a vector of six normalized variables representing the current and previous posted speed limit, speed, and density on the controlled urban motorway. The reward function was modeled as the proportional negative $TTS$ measured between the two control time steps. Oscillations between consecutive speed limit changes were prevented by restricting them to 20 km/h. A QL-based VSL (QL-VSL) control approach was also proposed in [46] to reduce $TTT$ at motorway bottlenecks. The QL-VSL approach significantly outperformed the feedback-based VSL strategy based on the obtained results. In [47], the Reinforcement-Markov Average Reward Technique (R-MART) approach was proposed for VSL. The long-term middle rewards are utilized in the R-MART algorithm opposed to QL algorithm deferred rewards. The application of the R-MART VSL controller was able to reduce $TTS$ by 18% and almost 20% less $CO_2$ emissions compared to the case without VSL. In our previous study [48], we proposed a deep QL-based VSL algorithm including a customized learning process and a complex reward function consisting of three separate objectives. The proposed algorithm aims to increase the throughput of the motorway by increasing the average mainline speed, increase safety by decreasing the difference between measured speed and posted speed limit, and minimize speed limit fluctuations in successive control step intervals. The proposed algorithm managed to increase the average mainline speed and reduce traffic density. The oscillations between the posted speed limits and the measured speeds were prevented.

The QL-VSL strategy was found to be most effective in reducing system $TTT$ in both stable traffic demand scenario and fluctuating traffic demand scenario. Improvements larger than 20% can be expected depending on the respective traffic situation.

### 2.3. Classical RM Approaches

As mentioned earlier, RM is used to control the traffic flow entering a motorway using traffic lights at the on-ramps. It was initially modeled as a pre-timed signal controller that was improved to operate on real-time traffic measurement data for traffic-responsive signal control. RM algorithms can be divided into localized single on-ramp controllers and system-wide or coordinated systems with multiple on-ramp controllers [49]. Most of the algorithms used recently employ a feedback control loop approach such as ALINEA. The detectors for RM feedback systems are shown in Figure 2.

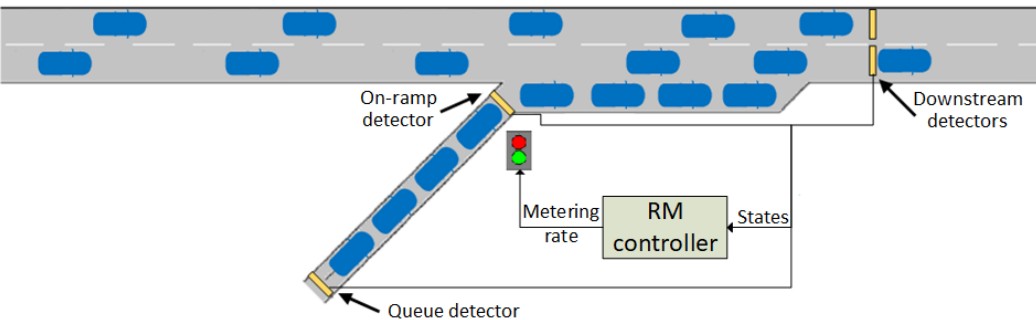

**Figure 2.** Block sheme of a general Ramp Metering (RM) controller.

The extensions of ALINEA use their own algorithms derived either for specific circumstances or as a new method for RM [19]. Coordinated RM has the main objective of controlling multiple upstream on-ramps before the on-ramp where the congestion occurs. Algorithms such as METALINE, Stratified Zone Metering (SZM), Heuristic Ramp-Metering Coordination (HERO), and System-Wide Adaptive Ramp Metering (SWARM) set the metering rates for each metered on-ramp. METALINE is the integral coordinated system version of ALINEA [50]. The SZM algorithm uses overlapping zones defined as motorway sections spreading between the two mainline detectors [51]. The commonly used length of an SZM zone ranges from 0.8 km to 4.8 km. The algorithm balances the sum of the upstream mainline flow, the sum of the unmetered on-ramp flow, the sum of the metered ramp flow, and the sum of the metered motorway flow with the sum of the off-ramp flow, the downstream bottleneck capacity, and the free capacity on the mainline [51].

HERO is an algorithm that uses a master–slave structure to control on-ramp traffic [52]. The master role is assigned to the bottleneck area on downstream on-ramp, while the slaves are assigned to the upstream on-ramps, and their on-ramp storage is used to solve the master on-ramp related congestion. When the algorithm is enabled, both the downstream and upstream metering rates are coordinated to keep the relative queue lengths approximately equal. The minimum queue length on the upstream on-ramp is updated as long as the queue length on the downstream ramp falls below the activation threshold.

The SWARM algorithm incorporates two independent algorithms. The first algorithm, called SWARM1, uses linear regression and Kalman filtering as well as a system-wide apportioning to forecast density, while the second algorithm, called SWARM2, is a local traffic-responsive system that converts measured densities into metering rates using linear conversion [53]. The more restrictive value defines the final metering rate.

### 2.4. RL-Based RM

An RM framework that uses RL as a robust and nonparametric generic optimization scheme that allows controlling of discretized partial differential equations was presented in [54]. An algorithm that enables experience sharing between agents and specialization of each agent due to multi-task learning to train Neural Networks (NNs), that was denoted from Mutual Weights Regularization (MWR), was also tested. MWR is an NN training approach that allows RL to learn a policy in a multi-agent environment without the curse of dimensionality due to the number of agents. Applied to RM's actual traffic control problem, a model-free approach achieves a comparable control level to the currently used

model-dependent implementation of the ALINEA algorithm. The results show that the average speed is globally increased while the total number of vehicles on the motorway is decreased. The MWR approach resulted in a significant performance improvement, almost reaching the performance of the ALINEA algorithm in both cases.

In [55], an algorithm based on the k-Nearest Neighbor Temporal Difference (kNN-TD) was developed and tested on a microscopic traffic simulation model, which was used as a replica of the real world and a test environment for evaluating three scenarios: do-nothing (no metering), ALINEA controller, and kNN-TD controller. The results show that the ALINEA controller reduced $TTT$ by 27%, while kNN-TD reduced $TTT$ by 44% compared to the no control scenario. The average queue length was reduced to 112 vehicles in the proposed method compared to 231 vehicles in the case of ALINEA controller.

In [56], an RL-based density Control Agent (RLCA) was proposed. The RLCA objective function is to optimize the mainline motorway density to minimize $TTT$ and maximize traffic flow. RLCA was tested with two different traffic network architectures and demand scenarios. In the first case study with a 6 km long mainline, the proposed RLCA guaranteed satisfactory performance by retaining the flow close to the motorway operational capacity. The simple QL algorithm managed to keep the mainline density close to 80 veh/km/2lanes, compared to the no control case where the density surpassed 110 veh/km/2lanes. The optimal control policy was determined based on two reference points: motorway capacity and $\rho_c$. In the second case study with a 500 m long mainline, RLCA always performed the optimal action.

In [57], a network-level RM framework was proposed based on modeling by the collaborative Markov Decision Process (MDP) and an associated cooperative QL algorithm based on a payoff propagation algorithm under the coordination graph framework. Three design strategies were analyzed. The first design was independent learning, where the greedy policy for agents was implemented, and an action was chosen to maximize its local reward. The second network-level design was fully distributed and based on the collaborative MDP and cooperative QL algorithm based on the payoff propagation algorithm. In this design, the actions were coordinated between the control agent and its neighboring agents. The cooperative Q-function was updated globally. The third design is a centralized cooperative multi-agent system that can be considered as a single large agent. The cooperative Q-function was updated with a single Q-function. The first design reduced the $TTT$ by 0.5%, the second by 6.5%, and the third by 6.9%.

The multi-agent Deep RL (DRL) RM algorithm, based on loop detectors data, was proposed in [58]. A multi-agent DRL framework is used to generate an appropriate RM scheme for each RM in real time to improve the operational efficiency of the urban motorway. Multi-agent proximal policy optimization architecture is introduced to solve the RM problem. The proposed algorithm was compared with no control, fixed-time control and ALINEA. Three traffic demand scenarios were tested which include constant demand, flat peak demand and sharp peak demand for both on-ramp and mainline traffic. The proposed algorithm increased the speed on the mainline by 43%, 29.7% and 11.5% for constant demand, flat peak demand and sharp peak demand scenarios, respectively. It can be seen that ALINEA achieved better results in the constant demand scenario, with a speed of 57.3 km/h on the mainline compared to 55.2 km/h for the DRL RM algorithm.

The kNN-TD method was able to achieve the best results, reducing the $TTT$ by 44% compared to the no control scenario, while the DRL RM algorithm increased the speed on the mainline by up to 43%. This emphasizes the motivation of using ML-based methods with the caveat that the increase in the effectiveness depends on the respective simulation scenario.

## 2.5. Impact of VSL and RM on HDV Traffic Flows

The first analysis of the impact of VSL on traffic flow dates back to 1972 [59]. This study was conducted with traffic data collected on German motorways. The same author extended the earlier analysis and concluded in [33] that VSL significantly contributes to the

homogenization of vehicle speeds in traffic when VSL is applied to the light or medium traffic demand. In this case, VSL consequently causes a decrease in the average speed of the traffic flow and leads to speed homogenization. Based on the results of [59], study [60] provides a quantitative description of the impact of VSL on the fundamental flow–density diagram. A model of the basic diagram was constructed, showing that as the amount of the speed limit is reduced, the capacity of the fundamental diagram increases, i.e., there is an increase in the vehicle flow. Such an increase in capacity was later found to be inaccurate. Research in [61] could not find any capacity increase attributable to the VSL application. In [62], the analysis on the effectiveness of the VSL in relation to the vehicle speed was carried out on 22 sections of national roads in Poland. The research included 14 rural sections with legal speed limits of 90 km/h, 70 km/h and 50 km/h, and eight sections of national roads with a VSL system installed with 90 km/h and 50 km/h speed limits. The analysis showed a very positive impact of such a system, resulting in an effective and significant reduction in average speeds and the number of noncomplying vehicles, as well as improved speed harmonization.

In [63], the impact of the fixed time RM and ALINEA RM on peak hour traffic flow was analyzed and compared. The results show that RM has no significant effect on traffic flow at average demand, but at high demand, the delay time of vehicles in this segment was reduced by 91% and 64.8% for the fixed time RM and ALINEA algorithm, respectively. At low demand, the delay time was increased both on the motorway and on the on-ramp, thus proving the RM is ineffective in such scenarios. It is noted that the RM has positive effects on traffic operations and reliability, but it could create a new (possibly "hidden") bottleneck to occur downstream, thus diluting the overall benefits [64]. In [64], ALINEA and HERO RM strategies were tested. ALINEA produced a better *TT* than HERO with smaller differences between the two. The throughput and congestion duration were also improved at all congestion levels. In the following subsections, the surveyed algorithms are grouped according to common metrics and performance indicators of the algorithm, such as safety, emissions, and macroscopic traffic parameters. First, the impact of VSL is analyzed, and then the impact of RM is added into the analysis.

### 2.5.1. VSL Impact on Stable Traffic Flows

According to [7], a stable traffic flow is characterized by the fact that the traffic flow density $\rho$ (veh/km) is less than $\rho_c$ (left green side in the approximated triangular fundamental diagram, Figure 3). In a stable flow, traffic runs smoothly without much interaction between vehicles. The impact of VSL on a stable flow is evident from the reduction in mean speed. After the flow adapted to the newly set VSL speed limit, it remained stable, but at a lower mean speed and higher density compared to VSL-free traffic flow. VSL application is characterized by a lower than $\rho_c$ value resulting in an increased *TT* and a reduced LoS.

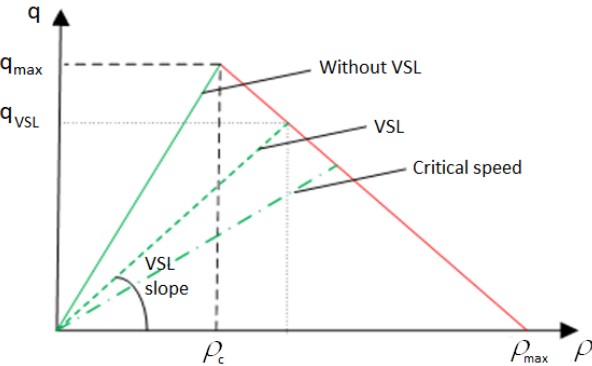

**Figure 3.** Impact of VSL on triangular fundamental flow–density traffic diagram [6].

The temporary decrease in vehicle flow due to the transition of traffic flow state after the start of VSL operation is due to the fact that the flow density is higher in the

state under the effect of VSL compared to uncontrolled traffic flow (lower speeds mean less space between vehicles). Due to the change in flow density, the flow is temporarily reduced, and a new higher density is created (active VSL). This influence of VSL on traffic flow was determined in [2]. The application of VSL upstream of a bottleneck that is in a state close to the occurrence of congestion can temporarily reduce the flow of vehicles approaching the bottleneck area. This potentially relieves the bottleneck area and delays or even prevents congestion.

### 2.5.2. VSL Impact on Unstable Traffic Flows

When the traffic flow density reaches values above the critical density ($\rho > \rho_c$), the traffic flow becomes unstable (right red side in the approximated triangular fundamental diagram, Figure 3). Interactions between vehicles are then more pronounced. The resulting traffic disruption (caused by, e.g., vehicle braking in a platoon) creates a shock wave that triggers a chain reaction that can produce a complete traffic jam on the urban motorway. Complementing the previous findings, more detailed studies on the effects of VSL on unstable traffic flow are summarized in [6,65]. It can be seen from [33] that the $\rho_c$ value in the fundamental diagram shifts to a higher value due to the effect of VSL (Figure 3). Therefore, it is possible to place more vehicles on the same length of the motorway segment without the traffic flow going into an unstable condition. This effect was confirmed in the study [6] and is attributed to the effect of speed homogenization mentioned earlier. Regarding the capacity increase, the research results are insufficient, as there is a slight capacity increase in certain locations of the observed urban motorways.

### 2.5.3. Impact of VSL and RM on Traffic Safety and Emissions

The VSL system can monitor the propagation of lower speeds in the traffic flow opposite to the direction of traffic on the motorway caused, for example, by a traffic incident or an active bottleneck, and set a new speed limit to allow the arriving vehicles to gradually adjust their speed to the new traffic situation [66]. Safety improvement was observed in [30,67,68] with appropriate VSL actions that prevent sudden changes in vehicle speeds. A study in [69] confirmed an increase in safety on the M25 motorway in the United Kingdom, while study [70] found increased safety on the E4 motorway in Stockholm, Sweden. A significant reduction in accident risk when VSL was applied in free-flow traffic was observed in [29]. In contrast, no significant improvement was observed in increased traffic flow. As a consequence of the reduction in the speed limit, the application of VSL reduces the average speed of the vehicle and, therefore, the resistance to the movement of the vehicle (most significant is the air resistance, which is proportional to the square of the speed). This reduces the fuel consumption required to overcome the resistance to vehicle movement, and hence the harmful emissions of exhaust gases. In [68,71–73], analyses have shown that the application of VSL can reduce emissions and fuel consumption by 4–6%.

In [74], a local RM control strategy, PI-ALINEA, was proposed. Four scenarios were formulated based on congestion levels, where the first scenario has the biggest congestion, while scenario four is close to the free-flow condition, corresponding to decreasing congestion. The total emissions for the first three scenarios are reduced by 16.9%, 6.7% and 4.7%, respectively, for the RM queue length constraint of 100 vehicles. The total emissions for the first three scenarios are reduced by 12.9%, 7%, and 5.3%, respectively, for the RM queue length without constraint. In [75], RM was adopted to analyze changes in traffic flow and associated variations in $CO_2$ emissions. The simulation results show a reduction in $CO_2$ emissions of 818.4 kg/h. The effectiveness of RM was reduced by quantifying a decrease in $CO_2$ emissions by 7.3%. RM reduced the $CO_2$ emissions to 3273.6 kg/day and 1194.9 tons/year.

## 3. Research and Implementation

### 3.1. Defined Research Questions

In this study, we focused on VSL and merging control approaches in mixed traffic flows on motorways. We reviewed papers that proposed methods for VSL and merging control in mixed traffic flow scenarios. To achieve our objectives, we formulated four main research questions:

- RQ1: What is the impact of AVs and CAVs on the fundamental diagram of mixed traffic flows?
- RQ2: How can the existing methods for VSL and RM be used to control mixed traffic flows?
- RQ3: What types of control algorithms have been proposed for VSL and merging control for mixed traffic flows?
- RQ4: What are the current open problems and what are the prospective research directions?

### 3.2. Applied Research Method

We focused on articles available online and published in English between January 2010 and 31 October 2020. We included the following digital libraries:

- Scopus;
- IEEE;
- Web of Science (WoS).

We used keyword-based searches to identify primary studies to filter relevant articles based on appropriately selected keywords (variable speed limit, ramp metering, machine learning, reinforcement learning, autonomous vehicles, connected and autonomous vehicles) used in the WoS digital library as an example. Table 1 shows the most representative papers on AV and CAV impact on fundamental diagram. Tables 2 and 3 show the most representative approaches for VSL and merging control in mixed traffic flows, with the proposed approaches and the obtained results. More details are given in continuation.

**Table 1.** Influence of Autonomous Vehicles (AVs) and Connected AVs (CAVs) on fundamental diagram and macroscopic traffic parameters.

| Paper | Year | Penetration Rate | Obtained Influence |
|-------|------|------------------|--------------------|
| [76] | 2020 | 0%–70% AVs and CAVs | reduced congestion and conflicts |
| [77] | 2019 | 0%–100% CAVs | increased flow, reduced acceleration and speed oscilations |
| [78] | 2018 | 0%–100% AVs and CAVs | increased capacity |
| [79] | 2018 | 0%–100% AVs | 48% increased $\rho_c$ value |
| [80] | 2017 | 0%–70% CAVs | 60% increased $\rho_c$ value 71% increased flow |
| [81] | 2017 | 0%–100% AVs | 32% increased free-flow speed, 29% increased flow under $\rho_c$ |
| [82] | 2016 | 0%–100% AVs and CVs | increased throughput |
| [83] | 2016 | 100% AVs | 43.6% increased flow |

**Table 2.** The most representative frameworks for VSL in mixed traffic flows.

| Paper | Year | Control Strategy | Proposed Method | Penetration Rate | Compared with | Improvements |
|-------|------|------------------|-----------------|------------------|---------------|--------------|
| [84] | 2020 | DVSL | DRL | 100% CAVs | no control, QL, DQN VSL, VSL-AC | 8.1% lower $TT$, 3.7% lower $CO$ emissions |
| [85] | 2020 | VSL and LC | GA | 100% CAVs | no control, VSL | 45% lower $TTS$ |
| [86] | 2020 | VSL | Multiclass CTM | 0%–100% AVs | no control | 33% lower energy consumption |
| [87] | 2019 | Speed harmonization | Optimal control with Hamiltonian function | 100% AVs | no control, VSL, SPD-HARM | 22% lower fuel consumption, 30% lower $TT$ |
| [88] | 2019 | VSL | GA | 0%–10% CAVs | 7 scenarios with no control, VSL, CTM and I2V with V2I | 36% lower $TTT$, 68% lower delays, 66% lower number of stops, 7.6% lower emissions |
| [89] | 2019 | VSL | Rule-based VSL | 0%–100% AVs | no control | 26% lower $MTT$, 31% lower fuel consumption |
| [90] | 2018 | Centralized VSL | Deep-RL (GRU) | 100% AVs | no control, feedback RM | 25% higher bottleneck throughput |
| [91] | 2017 | ACC and VSL | Rule-based VSL | 0%–100% AVs | ACC only, VSL only | 80% lower $TIT$, 77% lower $TET$ |
| [92] | 2016 | C-VSL | Feedback control | 0%–100% AVs | P-VSL | 49.5% lower delay time |
| [93] | 2015 | VSL | MPC | 50% and 100% CAVs | no control | 20% lower $TTT$, 11% improved safety, 16% lower fuel consumption |

**Table 3.** The most representative frameworks for merging control in mixed traffic flows.

| Paper | Year | Control Strategy | Proposed Method | Penetration Rate | Compared with | Improvements |
|-------|------|------------------|-----------------|------------------|---------------|--------------|
| [94] | 2019 | LC, merging | rule-based CLCC, event-based CMC | 100% CAVs | no control | 95% lower total delay time, 52% increased speed |
| [95] | 2019 | merging | trajectory planning | 100% CAVs | no control | achieved desired merging speed |
| [96] | 2019 | LC | decentralized and centralized control | 0%–100% AVs | no control, ALINEA RM | 43%–61% lower $TTT$ |
| [97] | 2019 | RM | look-ahead cruise control | 20% AVs | - | queue length optimization |
| [98] | 2018 | merging | V2X communication | 100% CAVs | no control | 5.3% lower $ATT$, 3.4% higher average speed |
| [99] | 2017 | merging | CIDM safe time gaps | 0%–25% AVs | - | 72% lower speed oscilations, 15.4% lower $ATT$ |
| [100] | 2017 | merging | Hamiltonian analysis | 100% CAVs | no control | 48% lower fuel consumption, 13.5% lower $TTT$ |
| [101] | 2017 | merging | nonlinear optimization | 100% CAVs | no control, gradual speed limit reduction | reduced delay, increased average speed up to 95 km/h |

### 3.3. Impact of AVs and CAVs on the Fundamental Diagram

As mentioned, Table 1 summarizes obtained research results on the impact of AVs and CAVs on fundamental diagram without control algorithms. In [76], an evaluation of the impact of AVs on an existing motorway, national road network, and urban network (Dublin city center) was presented. AVs were formulated as Society of Automotive Engineers (SAE) level of automation 2, and CAVs were formulated as SAE level of automation 4 with

Cooperative Adaptive Cruise Control (CACC). Tested scenarios included penetration rates from 0% to 70% with mixed SAE levels of automation. CAVs showcased gradual improvement in safety and efficiency. Satisfactory and realistic results were observed at penetration rates of around 20% to 40%. The motorway scenario proved to be the one most affected by CAVs. At low CAV penetration rates, traffic congestion and conflict situations were increased in some regions of the motorway networks.

In [77], the flow–density relation, Time-to-Collision (*TTC*), acceleration rate distributions, and speed difference distributions were analyzed under different penetration rates of CAVs from 0% to 100% and with two different time headways. The total number of incidental situations in the mixed traffic flow under different CAV penetration rates was reduced. The traffic flow was significantly increased by about 2000 veh/h in the case of the desired time headway of 0.5 s, while in the case of 1.1 s, the increase was measured to be close to 500 veh/h. The results of acceleration rate distributions and speed difference distributions of the mixed traffic flow indicate that the increase in penetration rates of CAVs smoothed both acceleration and speed significantly.

In [78], lane capacity was analyzed for different AV and CAV penetration rates. Lane capacity was increased by 8.6% when the AV penetration was increased from 0% to 100%. When the CAV penetration rate was increased from 0% to 100%, lane capacity was increased by 188.2%. When the mixed traffic flow consists of CAVs and HDVs, the lane capacity increases approximately linearly from 2046 veh/h/lane to 6450 veh/h/lane as the CAV penetration rate increased from 0% to 100%.

In [79], the influence of AVs in mixed traffic flow and the influence of autonomous driving levels on the fundamental diagram were analyzed. The results show that as the AV penetration rate increased with higher automation level, the capacity of the overall network increased and the $\rho_c$ value on a single road was higher. The $\rho_c$ value was increased by almost 48% from no AVs to 100% AVs in mixed traffic flow.

In [80], Cellular Automata (CA) model was used to analyze the impact of CAVs on the fundamental diagram on a single-lane road. CA was used to partition the road into cells, where a vehicle may or may not be present in a cell. The results show that with 70% of CAVs, the $\rho_c$ value was increased by about 37%, while the capacity under the $\rho_c$ was increased by about 42%.

In [81], the impact of AVs with different penetration rates on the fundamental diagram was analyzed. As the AV penetration rate increased, the speed–density curve shifted to the right, resulting in a higher $\rho_c$ value. Therefore, increasing the AV penetration rate increased the mean speed at the same density, thus increasing the traffic flow. The free-flow speed was increased by about 34% with 100% AV penetration rate, while the increase in capacity under $\rho_c$ was measured to approximately 30%.

In [82], the impact on traffic flow and density of Connected Vehicles (CVs) and AVs was analyzed under different penetration rates from 0% to 100% and three scenarios. In the first scenario, the effect of CV penetration rate was tested. The mainline flow was set to 1800 veh/h/lane in the first and second set of simulations. The throughput was increased as the CV and AV penetration rates increased. At low penetration rates of 0% to 50%, the results on the relationship between flow and density were inconsistent. At higher penetration rates, no breakdown or scatter was observed in the flow–density relationship. The second scenario analyzed the effects of AVs on throughput and scatter in the flow–density relationship. The results were similar to the first scenario, with the exception of a smaller scatter at 50% and 70% AV penetration rates. The third scenario analyzed the simultaneous effects of AVs and CVs on throughput and scatter in flow–density relation. Six combinations of different penetration rates of CVs and AVs were analyzed. The mainline flow was set to 2200 veh/h/lane. The scatter in flow–density relationship was increased once the number of CVs surpassed the number of AVs, except in the scenario with a high CV penetration rate.

In [83], another analysis of the impact of AVs on fundamental parameters ($q$, $\rho$ and $\bar{v}$) was performed. According to [83], the standard capacity values for a single lane of

2200 veh/h could increase traffic capacity to about 3900 veh/h in an 100% AV penetration rate scenario. This conclusion was made based on the changed headway time gap parameter for the following vehicle to 0.5 s. According to the authors [83], this very short following distance already occurs in up to 20% of all following distances, depending on the traffic conditions.

### 3.4. VSL in Mixed Traffic Flows

Table 2 demonstrates that VSL in mixed traffic flow scenarios has been broadly studied. In this subsection, we provide an overview of each study, starting with the earliest ones. To facilitate the evaluation of the surveyed methods, we grouped them according to their working concepts (optimization, rule-based, RL and others).

The studies [84,85,87–90,92,93,102–104] all analyzed the impacts of the proposed VSL control strategies in mixed traffic flows on macroscopic traffic parameters. These parameters include Mean Travel Time ($MTT$), $TT$, $TTT$, $TTS$, delays, number of stops, throughput, and speed. Although macroscopic traffic parameters give good insight into the efficiency of the algorithm, they are not the only criterion for measuring system performance. Studies [84,86–89,93,102] also evaluated the performance of the proposed strategies based on fuel consumption and vehicle emissions. Fuel consumption was measured in [87,89,93] while [84,88,102] measured vehicle emissions such as CO, $NO_x$ and HC. Proposed VSL control methods addressing safety were also analyzed as a control algorithm performance in [91,93]. Parameters including $TTC$, Time Integrated Time-to-collision ($TIT$) and Time Exposed Time-to-collision ($TET$) were used as safety measures.

#### 3.4.1. Optimization-Based Methods for VSL

An integrated VSL and Lane Change (LC) control framework for motorway bottlenecks with mixed traffic flows was proposed in [85]. The Model Predictive Control (MPC)-based framework considers the interaction between VSL and LC to maximize traffic efficiency. An improved multiclass Cell-Transmission Model (CTM) that considers the effects of LC behavior was proposed to predict the traffic state. The Genetic Algorithm (GA)-based method was used to find the optimal values of VSL and LC settings simultaneously. The simulations were performed under three different traffic demand values, 1000 veh/h/2lanes, 2000 veh/h/2lanes, and 3000 veh/h/2lanes with 100% CAV penetration rate. The proposed control method significantly reduced $TTS$ by 23.86% to 44.62%, outperforming the VSL-only control, which reduced $TTS$ by 6.43% to 13.84% compared to the no control scenario.

The problem of controlling the speed of a number of AVs before entering a speed reduction zone on a motorway was analyzed in [87]. The problem of deriving the optimal acceleration and deceleration of each AV was solved by the proposed optimal control method applying Hamiltonian analysis. The main objective was to minimize the acceleration or deceleration for each vehicle entering the controlled segment until the specified time. By minimizing the acceleration or deceleration of each vehicle, fuel consumption and emissions were reduced. The effectiveness of this approach was evaluated using three different traffic demand levels. It was compared to the no control HDV scenario, VSL algorithm (using shock wave theory proposed in [105]), and a modified vehicular-based SPeeD HARMonization (SPD-HARM) algorithm proposed in [106]. Fuel consumption for each vehicle was reduced by 19% to 22% compared to the no control HDV scenario, by 12% to 17% compared to the VSL algorithm, and by 18% to 34% compared to the SPD-HARM algorithm. $TT$ was improved by 26% to 30% compared to the baseline scenario, by 3% to 19% compared to the VSL algorithm, and 31% to 39% compared to the vehicular-based SPD-HARM algorithm.

In [88], the authors proposed an optimal VSL strategy in a CAV environment for a motorway corridor with multiple bottlenecks using an extended CTM that takes into account capacity drop and mixed traffic flow, including HDVs, heavy vehicles, and AVs. The GA optimization process for the developed VSL control strategy was conducted in

four steps. The first step involves the collection of motorway detector-based data in each cell, including traffic flow, speed, and density. The second step refers to the traffic state prediction in the selected motorway cells during the next control horizon based on the collected traffic data. The third step is the optimization process, which computes the objective function according to the predicted traffic data. The objective function value of each individual in the population is evaluated, and a new population is generated using the mutation and crossover operators. Two stopping criteria were used for the GA, including a maximum number of generations and the average relative change between the best objective function value at the current iteration and the values achieved up to the current iteration. The proposed VSL control with V2V, V2I, and I2V communication outperforms the VSL-only control. Better performance was achieved with increasing CAV penetration rate. Seven different simulation scenarios were compared with a scenario with 100% HDVs and no VSL control. The scenario with 10% CAVs, I2V, V2I, VSL control, and the extended CTM showed the best results, reducing $TTT$, delays, number of stops, and emissions by 35.6%, 67.9%, 65.9% and 7.6%, respectively.

In [92], AVs were used as actuators for Cooperative VSL (C-VSL), receiving speed limits directly from the controller and following them strictly. For a mixed traffic flow scenario, C-VSL was tested together with Point-level VSL (P-VSL), which was developed to inform HDVs that have a speed acceptance factor about the posted speed limit. Two scenarios were tested. The first contained only C-VSL, and the second used both C-VSL and P-VSL. In both scenarios, the AV penetration rate was changed, ranging from 0% to 100%. In the first scenario, the best results were obtained at a 40% AV penetration rate by reducing the delay by 49.5%. In the second scenario, the best results were obtained at a 90% AV penetration rate by reducing the delay by 47.9%.

An MPC-based VSL system was analyzed in [93]. In this approach, the future state is predicted and the control strategies are proactively updated in the system. The proposed MPC approach consists of four main components: data input and traffic state estimation, traffic state prediction, optimization using an objective function based on a rolling horizon, and a control action. The control time step used in this study was set to 1 min. A control horizon was used to account for the complexity and performance of the proposed system. The VSL control algorithm was based on a multi-objective function formulated with $TTT$ as a measure of network efficiency, $TTC$ as a measure of the instantaneous safety, the emission, and fuel consumption measures. This approach outperformed the scenario without VSL and resulted in $TTT$ reductions of about 20%, safety improvements of 6–11%, and overall fuel consumption reductions of 5–16% with 100% CAV penetration rate. The results also suggest that when the CAV penetration rate is 100%, optimization of only one component performs better. However, in scenarios with lower penetration rate, multi-objective optimization performs better in terms of mobility, safety, and sustainability simultaneously.

Another MPC strategy for VSL control of mixed traffic flows with CAVs was implemented in [104]. CAVs were modeled to have the same driving behavior as HDVs under no control condition and to fully comply with a set speed limit. A discrete first-order model that accounts for the capacity drop of shock waves was extended in this study to address the proposed VSL control. The proposed control structure of MPC consists of data processing and prediction based on traffic measurements, followed by the optimal control layer, control application layer, and intelligent motorway system. The used three-lane motorway model was 10 km long with a traffic demand of 5400 veh/h and a 20% CAV penetration rate. A disturbance speed lasting 2 min was generated to simulate shock waves, which was set to 5 km/h in the VSL-controlled segment. The free-flow speed was set to 100 km/h. The proposed strategy achieved total delay reduction of 3.7% from 35.3 veh · h to 34.0 veh · h.

### 3.4.2. Rule-Based VSL

In [89], a comprehensive evaluation of the potential impacts of AVs on an existing motorway system was analyzed in various motorway operational scenarios, including

heavily congested traffic (>95% of motorway capacity), lightly congested traffic (≈70% of motorway capacity), free-flow traffic (≈50% of motorway capacity), and future traffic (three times more than heavily congested traffic volume) conditions. A number of MoEs were measured to reflect the changes in mobility, safety, fuel consumption and emissions caused by the deployment of AVs [89]. The rule-based VSL approach was modified to mixed traffic flow conditions and also tested in different AV penetration rates. These conditions were classified as either free-flow, light congestion, or heavy congestion. The VSL control step time was set to 5 min. The minimum time-gap was set to 0.5 s for AVs and 1.1 s for HDVs. The driver imperfection was set to 0% and 50% for AVs and HDVs, respectively. The acceptance factor for speed limits was set at 1% for AVs and 15% for HDVs. For all traffic congestion scenarios, AVs showed improvements for all MoEs, as did VSL in mixed traffic flow scenarios. For heavy traffic congestion scenario, VSL improved $MTT$ by 14% to 26% at AV penetration rates ranging from 0% to 70%. AVs improved $MTT$ by 43% at AV penetration rates from 0% to 100% in the no control scenario. Fuel consumption was improved by 5% to 44% in the no control scenario at AV penetration rates from 0% to 100%. The case with VSL improved fuel consumption by 10% to 31% at AV penetration rates from 0% to 70%.

The combination of Adaptive Cruise Control (ACC) equipped SAE level 2 of driving automation AVs and VSL on the safety on s congested motorway was analyzed in [91]. The car-following Intelligent Driver Model (IDM) was calibrated for the simulation framework. The tested segment was 10 km long and contained 10 evenly spaced loop detectors. The mainline traffic demand was set to 1600 veh/h/lane, and a bottleneck was set to occur after 10 min between two consecutive detectors. $TET$ and $TIT$ parameters were used to assess the safety performance of the proposed method. $TIT$ was decreased by 57.6% to 79.9% from 0% to 100% AVs in the CACC and VSL model compared with ACC-only and VSL-only model which decreased it by 34% to 76.6% and 28.8%, respectively. $TET$ was decreased by 48.1% to 77.4% from 0% to 100% AVs in the CACC and VSL model compared to ACC-only and VSL-only models that decreased it by 39.7% to 67.8% and by 27.1%, respectively.

### 3.4.3. RL-Based VSL

In [84], a DRL model was proposed for Differential Variable Speed Limit (DVSL) control where dynamic and distinct speed limits can be imposed among lanes. It is based on the Actor-Critic (AC) architecture to learn a large number of discrete speed limits in the continuous action space. Four reward scenarios were tested. The reward scenarios $r_1$ and $r_4$ are based on several factors such as on-ramp inflow and off-ramp outflow. The reward scenario $r_2$ was the average speed in the downstream bottleneck. The reward scenario $r_3$ was computed by the total number of emerging braking vehicles in the controlled section. DVSL-$r_1$, DVSL-$r_2$, and DVSL-$r_4$ methods showed Average Travel Time ($ATT$) reduction of 1.08%, 4.1%, and 2.94%, respectively, while DVSL-$r_3$ showed an increase in $ATT$ of 6.2%. DVSL-$r_2$ had the best emission reduction in $HC$ and $No_x$, while DVSL-$r_4$ showed the best emission reduction in CO and PMx. The AC-based DVSL agent was compared to the baseline scenario with no control, and baseline scenarios with AC and Deep Q-Network (DQN)-based VSL controllers. The DVSL improved $ATT$ by 8.1% in scenarios with incidents and 5.8% in scenarios without incidents, while the VSL-DQN improved $ATT$ by 1.7% and 2.9%, respectively. The VSL-DQN and QL performed better than the VSL-AC by a very slight margin.

In [90], a traffic flow control system was tested that includes AVs whose speeds are controlled by a centralized agent based on deep learning. The learning structure is based on Recurrent Neural Networks (RNNs), where the Gated Recurrent Unit (GRU) algorithm solves the vanishing gradient problem that occurs when learning a classical RNN network. Introducing 10% AVs into the traffic flow enables better control of the traffic flow in a structure where each AV is a mobile actuator of the control system. The state of the agent's environment is determined by the density and average speed of the HDVs and the AVs.

In both cases, the data are measured for each lane on each model segment. The value of the outflow measured at the exit of the bottleneck was also taken into account to describe the state. The actions were defined by the speed selection for AV vehicles. The goal of the agent is to maximize the outflow from the bottleneck. Therefore, the reward function is formulated as the output flow from the bottleneck area, measured at intervals of 20 s. Compared to the case without control, the results showed that the AV controller for medium traffic demand (inflow $\geq$ 1800 veh/h) achieves a $\approx$25% higher outflow. For the lower traffic demand scenario, the case without control showed a better result. The results compared to the case where the mainline is controlled by RM (control feedback) showed that the AV controller provides approximately the same outflow from the bottleneck for increased traffic demand (inflow $\geq$ 1800 veh/h) as RM. The case with RM showed a much better result for lower traffic demands.

### 3.4.4. Other VSL Approaches

In [86], the flow–density relationship of moving bottlenecks with mixed traffic flow was analyzed under the influence of VSL based on the trapezoidal flow–density relationship aimed at reducing energy consumption. The algorithm was tested using CTM. The results show that in a demand scenario of 1800 veh/h, the total energy savings increased as the compliant vehicles' penetration rate increased. VSL reduced the energy consumption of compliant vehicles by up to 33% and by up to 8% for non-compliant vehicles. The total energy savings are more significant in a low demand scenario with a demand of 1800 veh/h. The results indicate that the free-flow speed does not have substantial effects on the percentage of energy savings.

In [102], the influence of Infrastructure-to-Vehicle (I2V) communication, AV control, and individualized speed limits for VSL by proposing a C-VSL extension was analyzed and compared with the classical VSL strategy. The I2V communication, AV control and individualized speed limits contributed to the harmonization of traffic flow and the reduction in exhaust emissions. C-VSL was implemented as VMSs functioning as roadside units, sending individual speed limit information to vehicles via I2V communication. Speed limits were calculated based on the distance between the VMS and the vehicle, the current speed of the vehicle, and the reference speed displayed on the VMS. Speed harmonization was achieved by giving individual speed limits to each vehicle at predefined time intervals. The proposed C-VSL narrowed the acceleration rate distribution and reduced $NO_x$ and $HC$ emissions. In the scenario with a C-VSL penetration rate of 30%, an increase in average speed was shown compared to a penetration rate of 100%.

In [103], CAVs were utilized as an alternative data source for a VSL control system. An interval type 2 fuzzy logic-based VSL system was proposed that employs CAVs to collect traffic data without the need for a fixed-point sensor. The performance of the proposed VSL system was evaluated and modeled in the microscopic simulator using a real motorway section located in Auckland, New Zealand. The results show that the proposed VSL system has similar performance to the detector-based system when more than 10% of CAVs are deployed. However, it is noted that VSL may become obsolete at very high CAV penetration rates.

### 3.5. Merging Control Approaches in Mixed Traffic Flows

Studies listed in Table 3 have shown that merging control approaches have also been extensively studied in mixed traffic flow scenarios. The majority of papers that analyzed merging control approaches evaluated the proposed algorithms in terms of macroscopic traffic parameters, such as speed, throughput, delay, on-ramp queue length, *TTT*, and *ATT* [94–101,107]. On the other hand, a few papers analyzed in this review evaluated the success of the proposed merging control approaches without considering the impact on the traffic flow macroscopic parameters [108–110]. In this subsection, an overview of each study is provided. Again, all surveyed studies are grouped according to the applied methods (optimization, car-following and trajectory planning, RL, and others).

### 3.5.1. Optimization-Based Methods for Merging Control

The problem of optimal coordination of CAVs at merging roadways formulated as an unconstrained optimal control problem by applying the Hamiltonian analysis to derive an analytical, closed-form solution was analyzed in [100]. Different scenarios were tested regarding mainline and on-ramp speed and density. The overall cumulative fuel consumption with imposed control was improved by 48.1% for the case with different initial speeds for each road compared to the baseline scenario. The *TTT* was also improved by 13.5%.

In [101], the authors proposed an optimization framework and an analytical closed-form solution for online coordination of CAVs at on-ramp merging zones formulated as a nonlinear optimization problem. At a distance of 250 m to 500 m before the merging area, the optimized acceleration information for each vehicle is sent back to the related vehicles at the next decision interval (10 s intervals), which strictly apply the given information. This approach was compared with the gradual speed limit by reducing the speeds to 50 km/h and 70 km/h at 0–250 m and 250–500 m before the merging area, and with no control scenario. Low (300 veh/h), medium (500 veh/h), and high (700 veh/h) demand scenarios for on-ramp and mainline flow were tested. The proposed approach reduced the average delay time per vehicle to almost 0 s and increased the average speed up to 95 km/h. For gradual speed limit and no control scenarios, the average delay time per vehicle under high on-ramp demand was measured to be up to 225 s and 255 s, respectively. The average speed was measured close to 15 km/h for both scenarios.

In [107], a hybrid approach was developed combining the hierarchical and distributed approaches for merging control. A Road Side Unit (RSU) was placed at the slot selection point and acted as a proxy between the mainline vehicles and the on-ramp vehicles. The RSU was utilized to locate the mainline vehicles and coordinate with them via V2V communication to determine suitable merging slots. These slots are marked as occupied for the vehicles and thus stored as a determined merging slot. Once the slot becomes free, the RSU distributes the slot information to the merging vehicle. The proposed approach does not suffer from system overload at very high traffic demand. Under medium mainline traffic conditions of 3600 veh/h, the slot-based approach with cooperation achieved a 106% increase in throughput compared to HDV-only flows. Under heavy mainline traffic conditions of 4700 veh/h, a throughput increase of 452% was measured for the slot-based driving with cooperation.

In [109], an MPC-based path planner was proposed to automatically decide the mode of maneuvers under a unified optimization framework for AVs in structured driving environments. A convex relaxation approach was used to determine the LC and lane-keeping maneuvers. Various simulation scenarios were tested in a simulation, including LC, lane keeping, ramp merging, and intersection crossing. The proposed path planner performance was effective in generating a safe and comfortable path for AVs in all tested scenarios.

In [110], a learning-based method for estimating vehicle intentions in ramp merging scenarios without over-the-air communication between vehicles was presented. The intention estimation is generated from a Probabilistic Graphical Model (PGM) that organizes historical data and latent intentions and determines predictions of real driving trajectories to learn transition models. The PGM-based intention estimation is augmented with ACC model to generate appropriate acceleration and deceleration behavior. The proposed method performance was evaluated on real merging data as well as with a designed merging control in simulation. The proposed method had the lowest failure rate and improved intention estimation in the merging control. It is computationally efficient and does not require acceleration information about other vehicles.

### 3.5.2. Car-Following and Trajectory-Planning-Based Merging Control Methods

Authors in [95] developed a vehicle trajectory planning method for CAV coordination at on-ramps, formulating the planning tasks of the ramp vehicle and the mainline vehicle as two related distributed optimal problems. One simulation scenario with only three CAVs

was used. Results show that all the three facilitating vehicles entering through on-ramp were able to develop a suitable gap under the influence of their respective leading vehicles whose speeds are constantly changing. All three merging vehicles could maneuver to the desired merge locations with the desired merge-in speeds.

The traffic flow strategy with vehicles using an optimized look-ahead cruise control method for merging control was proposed in [97]. A balance was achieved between reducing the energy demand of traffic flow and simultaneously reducing the queue length in the controlled lanes. It was found that the proposed method has several advantages, such as simple implementation, handling system nonlinearities, and considering parameter dependencies [97].

In [99], the authors used the Cooperative IDM (CIDM) to examine the system performance under different AV penetration rates for motorway merging control. A proposed CIDM-based controller determines the acceleration and deceleration rate of AVs in response to the actions of surrounding vehicles to improve road capacity and stability. The results imply that CIDM-based AVs can eliminate or alleviate speed oscillations on the motorway. The results show that with increasing AV penetration rate and under different safe time gaps in the IDM with values 0.4, 0.6, 0.8., 1, and 1.2 s, the speed oscillations were reduced by 46.1%, 53.9%, 54.8%, 69.7%, and 72.3%, respectively. For the safe time gaps of 0.4, 0.6, and 0.8 s, *ATT* was reduced by 15.4%, 12.3%, and 12%, respectively, while for the other cases, *ATT* was increased by 3.9% and 5.12%, respectively.

### 3.5.3. RL-Based Merging Control

In [108], a merging control method was presented that allows AVs to merge into congested motorways using Multi-Policy Decision Making (MPDM) with passive Actor-Critic (pAC). The method does not require forward simulation as it uses estimated state values learned with pAC. The pAC uses the 3-vehicle systems to learn the merging control policy. Therefore, the ego-vehicle cannot always handle the condition changes due to the maneuvers for the adjacent vehicles. This approach was compared with quadratic programming and Z-learning. The pAC with the NN achieved a 97% success rate by combining an approximate nearest neighbor to mitigate the imbalance and sparsity of the data. The combined MPDM with pAC achieved a success rate of 92%, which is comparable to a real HDV merging decisions.

In [111], a DRL architecture for learning an on-ramp merging control policy was proposed. The long short-term memory architecture was implemented in the driving environment to account for the influence of historical and interactive driving behaviors. In the use case of ramp merging, the driver can selectively provide control inputs while the automated driving system tries to maximize the reward and achieve an effective merging maneuver [111].

### 3.5.4. Other Merging Control Approaches

In [98], the motorway on-ramp merging control system was proposed to take advantage of V2X communication. CAVs share information in a vehicular ad-hoc network through dedicated short-range communication. It is assumed that all vehicles are CAVs. The vehicle sequencing protocol is designed to arrange vehicles in a predefined sequence to cooperate with each other before merging, thus avoiding high collision risks and excessive energy consumption and pollutant emissions when reaching the merging area [98]. The merging control approach was divided into three procedures. The first procedure calculated the maximum possible speed of on-ramp vehicles, the second procedure calculated the estimated arrival time, and the third procedure assigned vehicle sequence identification. The proposed protocol improved the *ATT* and average speed by 5.3% and 3.4%, respectively, compared to the no control scenario.

A two-level LC control strategy utilizing capabilities of AVs to improve the traffic flow on motorways in the scenario of active bottlenecks at on-ramps and lane drops was analyzed in [96]. The first-level centralized control optimized traffic density across lanes

to balance traffic flow among lanes and reduced the number of LCs near the merge area. Second-level decentralized control mitigated merging conflicts due to localized LC. This strategy was designed to eliminate merging conflicts between AVs on the mainline and vehicles on the on-ramp and generate safe gaps for targeted LC for AVs. The proposed strategy improved traffic flow compared to the baseline scenario and ALINEA RM and reduced *TTT* and *TT* variations among vehicles traveling on the mainline and on-ramps. The proposed control outperformed ALINEA for base demand level and minor increases in demand, thereby increasing the operational capacity of the motorway.

In [94], an online merging control system for multi-lane motorway merging areas for a traffic flow with a 100% CAV penetration rate was presented, based on the optimization of the LCs and the trajectories of the following vehicles. A rule-based LC decision was used to balance the lane flow distribution before the merging area. A Cooperative LC Control (CLCC) optimization model was proposed to ensure safe and smooth LCs. Co-operative Merging Control (CMC) model was used for merging control near the merging area. A dynamic method with moving boundary point was proposed to coordinate the consecutive execution of the CLCC and CMC models. The CLCC model was based on a discrete-time linear system control model that optimized the longitudinal LC acceleration with the objective of maximizing the average speed. The CMC model assigned a merging sequence to each vehicle entering the cooperative merging area, and vehicle speeds were optimized. The simulations were conducted under four demand levels with three lane demand scenarios. The integrated CLCC and CMC results show a reduction in total delay time of 27% to 94.8% depending on the demand scenario. An increase in average speed was measured ranging from 2.7% to 52.2% depending on the demand scenario.

## 4. Discussion

The focus of the paper was on the description of VSL and merging controllers and proposed control algorithms in mixed traffic flow scenarios. We have identified several potential research directions to address the limitations of existing methods. The limitations are the need for an accurately estimated fundamental diagram and the implementation of cooperative VSL and merging control system in mixed traffic flows.

The main drawback of the efficiency of classical VSL algorithms in mixed traffic flows is their inability to adapt their control policy to a new traffic situation, in which they perform sub-optimally. In the last five years, according to Table 2, there has been an increase in the number of studies focused on improving and proposing new VSL control algorithms for mixed traffic flows.

The drawback of both VSL and merging control approaches in mixed traffic flows is the assumption that the communication network is error free and that there is no delay or loss of information sent to the vehicles. Although AVs and CAVs are still at an early stage of development, a comprehensive study that includes both the control approach and the communication network influence analysis is yet to be made.

In [80], an analysis of mixed traffic flow was performed with CA model on a road segment divided into 3000 cells. The length of each vehicle was set to 1 cell and the maximum desired speed was set to 5 cells/s. The model could be improved by adjusting the vehicle length and maximum desired speed, which would lead to more realistic scenarios and results by keeping the definition of cell lengths in mind. A successful DRL model for DVSL control, where dynamic and distinct speed limits can be imposed between lanes, was implemented in [84]. Four learning reward scenarios were tested. Even though the results are promising, a safety analysis on incidents and crash risk or collision rate should also be considered. Due to differential speed limits, LCs could potentially lead to unwanted incidents in real-world scenarios.

In [99], it was stated that CIDM system performance under different proportions of AVs for motorway merging should consider safety aspect for low safe time gaps. For mixed traffic flows, HDVs, unlike AVs, cannot maintain low safe time gaps, where HDVs could potentially have no space to perform safe LC when needed.

In [85], the authors successfully used GA to solve the MPC formulated problem of VSL and LC. Using the CAVs' information about their average speed, density, and flow rate in each cell, GA repeatedly generates a population of individual solutions. These individual solutions are evaluated based on the objective function value in the next prediction horizon according to the modified multi-class CTM with these collected data.

Most researches on merging control approaches in mixed traffic flows disregard the classical RM approaches. Instead of the classical RM control, which uses a traffic signal with real-time traffic data to determine the rate at which vehicles should enter the motorway, merging control approaches are proposed to resolve interactions and conflicts between vehicles in merging areas [95,98–101]. These strategies can be achieved by assuming that all vehicles are CAVs.

One of the major shortcomings of VSL and RM control algorithms for mixed traffic flows is that they can become obsolete at very high penetration rates. The advantages of AVs and CAVs in future mixed traffic flows at any penetration rate are pronounced in improving the macroscopic traffic parameters of urban motorways reducing the need for separate control.

The major shortcoming in the application of RL is the lack of a precise explanation of the learning process. The question of why the agent learned a particular strategy of action could not be answered accurately. Another issue that needs to be addressed is the cooperation of VSL and merging controllers in mixed traffic flows by applying control algorithms to improve MoEs and LoS.

In [94,95,98,100,101], the analyses of merging control approaches were performed under the assumption that all vehicles are CAVs. The analysis of different AV and CAV penetration rates and scenarios with mixed traffic flows would be beneficial since traffic is stochastic in nature. Additionally, it is more likely in the near future that there will be large variations and variances in AV and CAV penetration rates in mixed traffic flows. Therefore, the analyses of more realistic traffic scenarios would be useful.

## 5. Conclusions

The application of advanced control solutions in the domain of ITS enables further improvement of LoS on urban motorways. This paper reviews previous analyses of the impact of VSL and RM on traffic flow. The role of VSL and RM systems on urban motorways is described, and the operation of VSL controllers and RM control strategies is explained. Emphasis is placed on the description of VSL and RM control algorithms in mixed traffic flows. The impact of AVs and CAVs and the application of VSL and merging control approaches in mixed traffic flow scenarios are also analyzed. The results of recent papers on simulated traffic control systems for mixed traffic flows are also included in the review.

The observed open research areas of applying VSL and merging control approaches in mixed traffic flows on urban motorways are as follows. Establishing a multi-agent VSL and merging system based on agent cooperation to maximize system performance. The multi-agent approach could be used to utilize the moving zone area under the VSL, calculate the speed limit, and use AVs and CAVs as actuators for VSL and merging control systems on urban motorways. All previous analyses are either based on the analysis of only one system (VSL or merging control). The application of VSL and merging control in mixed traffic scenarios could potentially smooth out speed harmonization, increase safety at merging areas on on-ramps and increase overall motorway capacity due to the lower headway gap that AVs and CAVs utilize, and communication characteristics that allow CAVs to utilize the motorway capacity more efficiently with cooperative LC. Another open area of research lies in the analysis of weather conditions on the behavior of AVs and CAVs, which can affect the performance of the traffic control system including the in-vehicle placed low level driving systems of (C)AVs. The lack of such analysis represents a shortcoming in current studies of mixed traffic flows with both VSL and merging control approaches.

**Author Contributions:** The conceptualization of the study was conducted by F.V., K.K., D.Č. and E.I. The funding acquisition was conducted by K.K. and E.I. The writing of the original draft and

preparation of the paper was conducted by F.V. and K.K. All authors contributed to writing of the review and final editing. The supervision was conducted by E.I. Visualizations were conducted by F.V. All authors have read and agreed to the published version of the manuscript.

**Funding:** This work has been partly supported by the University of Zagreb and Faculty of Transport and Traffic Sciences under the grants "Control System of the Spatial-Temporal Variable Speed Limit in the Environment of Connected Vehicles" and "Innovative models and control strategies for intelligent mobility", by the Croatian Science Foundation under the project IP-2020-02-5042, and by the European Regional Development Fund under the grant KK.01.1.1.01.0009 (DATACROSS).

**Acknowledgments:** This research has also been carried out within the activities of the Centre of Research Excellence for Data Science and Cooperative Systems supported by the Ministry of Science and Education of the Republic of Croatia.

**Conflicts of Interest:** The authors declare no conflict of interest. The funding institutions had no role in the design of the study; in the collection, analyses, or interpretation of data; in the writing of the manuscript, or in the decision to publish the results.

## Abbreviations

The following abbreviations are used in this manuscript:

| | |
|---|---|
| AC | Actor-Critic |
| ACC | Adaptive Cruise Control |
| AV | Autonomous Vehicle |
| *ATT* | Average Travel Time |
| CA | Cellular Automata |
| CACC | Cooperative Adaptive Cruise Control |
| CAV | Connected and Autonomous Vehicle |
| CIDM | Cooperative Intelligent Driver Model |
| CLCC | Cooperative Lane Changing Control |
| CMC | Cooperative Merging Control |
| CTM | Cell-Transmission Model |
| CV | Connected Vehicle |
| C-VSL | Cooperative Variable Speed Limit |
| DVSL | Differential Variable Speed Limit |
| DRL | Deep Reinforcement Learning |
| DQN | Deep Q-Network |
| GA | Genetic Algorithm |
| GRU | Gated Recurrent Unit |
| HDV | Human-Driven Vehicle |
| HERO | Heuristic Ramp-metering Coordination |
| IDM | Intelligent Driver Model |
| ITS | Intelligent Transport Systems |
| I2V | Infrastructure-to-Vehicle |
| kNN-TD | k-Nearest Neighbor Temporal Difference |
| LC | Lane Change |
| MDP | Markov Decision Process |
| ML | Machine Learning |
| MoE | Measure of Effectiveness |
| MPC | Model Predictive Control |
| MPDM | Multi-Policy Decision Making |
| *MTT* | Mean Travel Time |
| MWR | Mutual Weights Regularization |
| NN | Neural Network |
| pAC | passive Actor-Critic |
| PGM | Probabilistic Graphical Model |
| P-VSL | Point Variable Speed Limit |
| QL | Q-Learning |
| RM | Ramp Metering |

| | |
|---|---|
| RL | Reinforcement Learning |
| RNN | Recurrent Neural Network |
| RSU | Road Side Unit |
| R-MART | Reinforcement-Markov Average Reward Technique |
| RLCA | Reinforcement Learning Control Agent |
| SAE | Society of Automotive Engineers |
| SWARM | System-Wide Adaptive Ramp Metering |
| SZM | Stratified Zone Metering |
| SPD-HARM | SPeeD HARMonization |
| *TET* | Time Exposed Time-to-collision |
| *TIT* | Time Integrated Time-to-collision |
| *TT* | Travel Time |
| *TTC* | Time-to-Collision |
| *TTS* | Total Time Spent |
| *TTT* | Total Travel Time |
| VMS | Variable Message Sign |
| VSL | Variable Speed Limit |
| V2V | Vehicle-to-Vehicle |
| V2R | Vehicle-to-Roadside |
| V2I | Vehicle-to-Infrastructure |
| V2X | Vehicle-to-Everything |
| WoS | Web of Science |

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
