# Peer review of "Variable Speed Limit and Ramp Metering for Mixed Traffic Flows: A Review and Open Questions"

_applsci, doi:10.3390/app11062574_

Round 1

Reviewer 1 Report

Overall, the authors provide a well and comprehensive review of the various VSL techniques proposed by other researchers. The last column in tables 2 and 3 were especially useful at summarizing the merits of the different schemes. It would have been good to also include a quantitative comparison of the performance of the various algorithms presented in Section 2 in terms of a common set of metrics. Furthermore, it may be worthwhile grouping the different previous work in Sections 3.3 and beyond based on the type of methods, e.g. machine learning, optimization, etc.

Author Response

Dear reviewer,

Thank you for your time and comments that emphasize the positive parts of the paper and the parts that need to be improved. We have revised the paper, and with this letter, we report the point-by-point responses to your comments. Comments are labeled with bold letters and the response with italic. All changes are labeled in the new version of the paper with red-colored words. An extensive English spell check is conducted.

(x) English language and style are fine/minor spell check required.

The spell check is conducted and applied to the original manuscript.

Overall, the authors provide a well and comprehensive review of the various VSL techniques proposed by other researchers. The last column in tables 2 and 3 were especially useful at summarizing the merits of the different schemes.

Thank you for your comments.

It would have been good to also include a quantitative comparison of the performance of the various algorithms presented in Section 2 in terms of a common set of metrics.

The quantitative comparison of the performance of the various algorithms is summarized in subsections of Section 2 (lines 334-427). We added the description of the common set of metrics used in section 2 (lines 363-367). We summarized the best RL method used for improving macroscopic traffic parameters (lines 236-239 and 329-333).

Furthermore, it may be worthwhile grouping the different previous work in Sections 3.3 and beyond based on the type of methods, e.g. machine learning, optimization, etc.

The papers were grouped into subsections (lines 524-525) in section 3.4 (lines 538, 620, 652, 688) and 3.5 (lines 726-729, 780, 805, 822). Section 3.3 was not grouped by types of methods since it only reviews the papers on the impact of AVs and CAVs without control algorithms as described in lines 456 and 457.

Once again, thank you for your comments.

We look forward to hearing from you at your earliest convenience.

Reviewer 2 Report

The subject of the article is topical and important from the point of view of the ITS modeling domain. The article may be published as is. 

Author Response

Dear reviewer,

Thank you for your time and comments that emphasize the positive parts of the paper and the parts that need to be improved. We have revised the paper, and with this letter, we report the point-by-point responses to your comments. Comments are labeled with bold letters and the response with italic. All changes are labeled in the new version of the paper with red-colored words. An extensive English spell check is conducted.

The subject of the article is topical and important from the point of view of the ITS modeling domain. The article may be published as is.

Thank you for your comments.

Once again, thank you for your comments.

We look forward to hearing from you at your earliest convenience.

Reviewer 3 Report

I thoroughly enjoyed reading and reviewing the present manuscript. The authors present a comprehensive survey of Variable speed limit and Ramp metering control algorithms for mixed traffic flow.

Firstly, the writing of this manuscript is very well. However, I will perform some comments to improve this manuscript presented by the authors:

  1. At the end of the introduction section, It is better to provide a paragraph about the organization of this manuscript.
  2. Although the authors provide a comprehensive survey about VSL and RM in mixed traffic flow, I would like to see a thorough analysis of the impact of VSL and RM with current developed situations.
  3. It is better to present a more deep discussion on the limitation of VSL and RM algorithms for mixed traffic flows.

Author Response

Dear reviewer,

Thank you for your time and comments that emphasize the positive parts of the paper and the parts that need to be improved. We have revised the paper, and with this letter, we report the point-by-point responses to your comments. Comments are labeled with bold letters and the response with italic. All changes are labeled in the new version of the paper with red-colored words. An extensive English spell check is conducted.

(x) English language and style are fine/minor spell check required.

The spell check is conducted and applied to the original manuscript.

I thoroughly enjoyed reading and reviewing the present manuscript. The authors present a comprehensive survey of Variable speed limit and Ramp metering control algorithms for mixed traffic flow.

The writing of this manuscript is very well.

Thank you for your comments.

At the end of the introduction section, it is better to provide a paragraph about the organization of this manuscript.

The paragraph about the organization of this manuscript is presented at the end of the introduction section (lines 131 - 137).

Although the authors provide a comprehensive survey about VSL and RM in mixed traffic flow, I would like to see a thorough analysis of the impact of VSL and RM with current developed situations.

As mentioned, the focus of this review paper was on a survey of VSL and RM control strategies in mixed traffic flows. We did a short survey on the current methods used on classic traffic flows containing only human-driven vehicles (HDV) in section 2 (lines 138-427). In that section, we mention key measures of the effectiveness of the proposed methods such as safety, emissions, and macroscopic traffic parameters such as Total travel time, Total time spent, and Travel time including an impact description on HDV traffic flows (subsection 2.5). We did mention other review papers that focused on VSL and RM control strategies for classic traffic flows containing only human-driven vehicles (lines 112 – 116). Since the review papers on this subject already exist, we did not go in-depth with the methods used for classic traffic flows and just emphasized that subsection 2.5 contains the impact analysis on HDV traffic flows. If this does not address the needed improvement, we would be grateful for some more detailed guidelines for this improvement.

It is better to present a more deep discussion on the limitation of VSL and RM algorithms for mixed traffic flows.

A deeper discussion on the limitation of VSL and RM algorithms for mixed traffic flows is added in section 4 (lines 903-918).

Once again, thank you for your comments.

We look forward to hearing from you at your earliest convenience.
